# Protocol for a Delphi Consensus Study to Determine the Essential and Optional Ultrasound Skills for Medical Practitioners Working in District Hospitals in South Africa

**DOI:** 10.3390/ijerph19159640

**Published:** 2022-08-05

**Authors:** Pierre-Andre Mans, Parimalaranie Yogeswaran, Oladele Vincent Adeniyi

**Affiliations:** 1Department of Family Medicine, Cecilia Makiwane Hospital, Mdantsane, East London 5201, South Africa; 2Department of Family Medicine and Rural Health, Faculty of Health Sciences, Walter Sisulu University, Mthatha 5117, South Africa; 3Department of Family Medicine, Mthatha Regional Hospital, Mthatha 5100, South Africa

**Keywords:** Delphi method, district hospitals, family physicians, point of care ultrasound, South Africa

## Abstract

With increasing access to point of care ultrasound (POCUS) at district hospitals in South Africa, there is a lack of standardisation of skillsets among medical practitioners working at this level of care. This study protocol aims to use the Delphi process to achieve expert consensus on the essential and optional ultrasound skills required for medical practitioners working in district hospitals in South Africa. In alignment with the Delphi method, several iterative rounds will be implemented from June to November 2022. Purposive sampling will be conducted, through the recruitment of two representatives from each academic department of family medicine and two medical doctors working in district hospitals in each province in the country (N = 36). The POCUS skillsets published by the American Academy of Family Physicians will be circulated in the first iterative round, following which participants may suggest further additions. Once a consensus target of 70% has been achieved, the Delphi process will be finalised. The Delphi process and data analysis will be facilitated by an online Delphi platform. Findings from the study will provide insight into the design of the curriculum for POCUS training for medical practitioners in district hospitals and registrars in family medicine departments across the country.

## 1. Introduction

Point of care ultrasound is a phrase that was coined by emergency physicians in the 1990s. Better known as POCUS, it refers to a particular change in the approach to diagnostic ultrasound, where previously diagnostic ultrasound had been performed by taking the patient from the clinician to the radiology suite for a formal systematized diagnostic procedure. Ultrasound units have been transformed from the bulky units introduced to medicine in 1947, into the handheld smartphone-based applications that are both easy to use, and readily available. POCUS, at its heart, is about allowing the clinician to use diagnostic ultrasound as an extension of the clinical assessment; to have an ‘ultrasound stethoscope’ [1].

a.Benefits of point of care ultrasound (POCUS)

Of the diagnostic imaging modalities available, ultrasound is less expensive than computed tomography (CT) and magnetic resonance imaging (MRI) and offers no ionizing radiation risk [2]. In some contexts, ultrasound imaging may even be superior to other diagnostic modalities. One such context is the emergency department, where ultrasound outperforms chest X-rays in the evaluation of respiratory distress [3]. Ultrasound is also used as the preferred method for functional hemodynamic monitoring and differentiating between causes of shock [4,5].

Ultrasound improves the accuracy and reduces the complication rate for interventional procedures. This procedural benefit is so pronounced that ultrasound guidance is considered a minimum requirement for performing interventional procedures [6]. Since POCUS is a clinician-facilitated bedside test, it has been shown to reduce the time taken to diagnosis and discharge [7]. There is also evidence that bedside ultrasound directly improves the quality of the doctor–patient experience, enhances confidence in the doctor’s diagnosis and improves patient satisfaction [8].

b.Drawbacks of POCUS

Ultrasound is an operator-dependent examination, and some bedside operators may overestimate their skills, possibly causing increased patient morbidity [9]. Despite the initial excitement and massive uptake of POCUS, some studies have found that its use does not change clinical outcomes, only the time taken to obtain diagnosis of patients [9]. Prolonged exposure to ultrasound pressure waves can cause a change in tissue temperature, which may theoretically be harmful to the developing foetus, although no definite harm has yet been proven [2]. Since there is a myriad of clinical applications, the full scope of bedside ultrasound can easily exceed the capacity of the non-radiologist clinician. This has led to the necessary breakdown and structuring of specific ultrasound skillsets. The appropriate use of these POCUS skillsets pertains to specific contexts, where pre- and post-test probability for the specific population needs to be taken into account [7]. It is therefore not a quick, ‘one size fits all’ solution to any clinical dilemma but requires research validation for each condition and context.

POCUS also requires additional training, and available devices to use. This training and equipment availability is still a major barrier to its use, even in well-resourced contexts [9,10]. With increased use and evidence for the use of POCUS, the modality has unfortunately become another potential point for medical litigation. The fear of medical litigation has led to some specialties being reluctant to formally include POCUS in their curricula. However, current evidence is that cases of litigation are limited to situations where POCUS was not performed, or not performed in good time [9,11].

c.Current practice

Ultrasound units have become more affordable, mobile and user friendly. With more widespread use of ultrasound there has been an explosion in published research about its various applications. Examples of these are diverse and include the use of POCUS literally from head to toe—ultrasound is used to determine intracranial pressure through measuring diameter changes in the ocular nerve sheath [12], and for diagnosing plantar fasciitis [13], among many other uses.

The diverse use of ultrasound has mainly been developed in specialist centres, often by specialist radiologists. However, other fields of medicine have now followed with validation studies for most of the more common bedside applications. Examples of these range from family physician and general practitioner’s validation of POCUS cardiac, lung and aorta assessment [14] to nurse-led ultrasound assessment of fluid status in heart failure [15]. This variety of uses suggests that with minimal training, most medical professionals can safely use bedside ultrasound for specific indications in their context. It is however important to note that for each application and context, the specific validity needs to be proven.

Specialty-specific guidelines on what ultrasound skillsets are considered essential have also been developed for many subspecialties. These guide both the profession on what should be considered as core proficiencies, and the public on what examinations may be expected in the hospital setting. Examples of these include emergency medicine [16,17], internal medicine [18] and family medicine [8].

d.Limitations identified in current literature and practice

As each specialty has systematically developed its own ultrasound curriculum and guidelines, it has become apparent that differences not only exist among specialties, but also among geographical contexts. These can be seen in the difference between ultrasound use in rural Nepal, where the assessment of paediatric osteomyelitis is considered a basic skill, and its use in Rwanda, where the ultrasound diagnosis of HIVAN (HIV-associated nephropathy) and parasitic cysts are seen as basic skillsets [19,20]. This has led to significant variation in ultrasound use and curriculum content for the same specialty in different developed countries. As an example, the published ultrasound requirements for Scandinavian general practitioners are different from those for Slovenian general practitioners. The Scandinavian curriculum focuses largely on soft tissue ultrasound applications, which, though included, are not a focal point in the Slovenian curriculum. The Slovenian curriculum, in turn, regards the identification of renal stones with ultrasound as a core competency, while the Scandinavian curriculum does not [21,22,23]. If these differences exist among developed countries, they are likely to be even more pronounced among developing countries, and between the developing and developed world [23,24]. The developing world has a significantly different burden of disease, with a far greater focus on communicable diseases [25] than in the developed world. In addition, the developing world has limited resources for spending on healthcare and a different allocation of resources within healthcare [24,26,27,28].

There are no existing guidelines on ultrasound skills for medical practitioners in district hospitals in South Africa. Yet in the South African context, there is widespread availability and use of ultrasound at the district hospital level, as evidenced in its inclusion as a core element in the Ideal Hospital Manual [29]. The lack of guidance and formalised training in POCUS directly influences conformity and standards of service delivery at the district hospital level. This may have a negative impact not only on patient experience, but also on health outcomes [7,8,30,31].

## 2. Methods

The aim of this study is to elicit expert consensus as to the skills considered essential for point of care ultrasound (POCUS) for district hospital medical practitioners in the South African context.

The following objectives will provide guidance for implementation of the study:I.To determine the essential (mandatory) POCUS skills that all medical practitioners need in order to offer the required level of clinical services expected at district hospitals in South Africa.II.To determine the optional (relevant but geographically specific) POCUS skills that medical practitioners may require in order to optimally manage the geographically differentiated local burden of disease at some district hospitals in South Africa.

a.Study Design

To answer the question of what ultrasound skills a district hospital medical practitioner should possess, we will create a list of the appropriate ultrasound skillsets by establishing expert consensus. Previous publications on ultrasound curricula via expert consensus utilised either focus group discussions at specialist conferences, or the Delphi technique [17,32].

The Delphi technique was developed in the 1950s, as a means towards predicting technological advances in warfare via an expert consensus. Its name is derived from the oracle of Delphi, which was used in ancient Greece to predict the future. Despite its origins as a prediction tool, the Delphi technique has been extensively used in research as a means towards expert consensus.

The Delphi technique achieves consensus by progressively moving through sequential rounds of anonymous surveys with facilitated feedback after each round. At the end of each round, the facilitator reviews and collates all the participants’ responses. The facilitator then supplies each participant with a breakdown of the group’s responses on each aspect of the survey that did not achieve consensus in the previous round, as well as indicating which items did receive consensus. The facilitator will also moderate and include any comments made by participants regarding any aspect of the survey. This information helps participants to gauge their follow-up response to that of the rest of the group and thus facilitates consensus over multiple rounds. Participants’ responses and group feedbacks are always anonymous.

Responses may be binary or according to a Likert-like scale. As soon as consensus is achieved for a particular item of the survey, it is excluded from subsequent rounds. The percentage agreement required for consensus varies among studies, but is generally accepted at 70% [17,33,34]. We decided to use the Delphi technique because it is cost effective and allows geographically diverse participants to be included. In addition, it allows anonymous responses from participants to be collated.

b.Participants

To ensure that the final list of skillsets is appropriate for the geographic differences in the South African district hospital context, it is necessary to have adequate geographic representation. A purposive sampling approach will be used, but with specific precaution to limit selection bias. To ensure this, we will approach RUDASA (Rural Doctors Association of South Africa) to nominate two members from each province that have ultrasound proficiency and are frontline district hospital medical practitioners to participate in the study. Since there are nine provinces in South Africa, we will have 18 frontline medical practitioners. We will also request each HOD (Head of Department) from each academic family medicine department to nominate two participants from their department. These academic family physicians should have some experience in ultrasound use and play active roles in either the undergraduate or postgraduate programme in their respective universities.

Nelson Mandela University in the Eastern Cape does not yet have a well-established family medicine department and will not be included. Since there are nine family medicine departments, we will have 18 academic family physicians participating in this study as indicated in Table 1 [35]. This will bring the total number of participants to 36, a number which compares favourably with similar studies [3,17,22].

c.Starting questionnaire

As a starting point we will use the published POCUS skillsets of the American Academy of Family Physicians (Please see Participant Questionnaire under Appendix A) [8]. Each individual ultrasound skillset will be presented in point form and weighted by participants on a Likert-like scale. The scale will have three options. These are “essential” (mandatory for district hospitals), “optional” (geographically specific) and “non-essential” (not applicable for district hospital). During the first round, participants will be allowed to add any additional ultrasound skillsets to the list. These will be included in further iterations until consensus has been reached.

d.Implementation process

An email invitation with a participation link will be sent to each of the participants at the initiation of the study and at the beginning of each round. Weekly email reminders will be sent to those who have not responded, until either all participants have responded, or four weeks have passed. If a participant has not responded within four weeks, the principal investigator will contact them by telephone to determine if they prefer to withdraw from the study. Failure to respond after four emails and a telephone call will result in removal from the study. The responses will be tallied after each round, with feedback prepared and sent to each participant regarding the level of consensus reached by all participants in the previous round. All items that have reached 70% or more consensus will then be removed from further iterations of the survey. The Delphi process will continue until consensus has been achieved on all items.

In the event that there is no significant change in responses on a specific item (less than 5% change over more than two consecutive rounds), indicating that the required agreement for consensus (70%) cannot be reached, we can confidently say that consensus cannot be achieved for that specific item. The items on which consensus cannot be reached (less than 5% change over more than two consecutive rounds) will be removed from future iterations of the survey. At the end of each round, all skillsets that achieved 70% consensus or more will be removed from future iterations, until consensus has been achieved on all skillsets (as shown in Figure 1). This process towards consensus can take any number of rounds, but typically takes two to three rounds [36].

To ease this process, an online Delphi tool—Welphi (Lisbon, Portugal) will be used. This tool provides software to facilitate both data collection and management, but also automated reminders to participants. As part of the initial Delphi round, participants will also be asked to complete an online questionnaire documenting their demographics and working experience.

e.Study schedule

The study will be implemented according to the structure shown in Figure 2.

The literature review, research and development of the research proposal was completed over the second half of 2021 to the present. The proposal has been reviewed and approved by the Walter Sisulu University Research, Ethics and Biosafety Committee. Recruitment of participants via both RUDASA (Rural Doctor’s Association of South Africa) and the various universities academic head of family medicine has commenced. It is our intent to start the online Delphi process in July 2022. This study is dependent on participant responses, and response times may be a major influencing factor for the duration of the study. There will be maximum of two months’ duration per round of the Delphi process, with an estimated maximum of four rounds required for consensus. This could mean a maximum of eight months for the data collection, starting July 2022, through February 2023. Once the data collection process is complete, we will require, at most, six months to finalise data analysis and write up.

f.Ethical considerations

Participants will be nominated by either RUDASA or their respective academic HOD’s, but participation will be completely voluntary. Each participant will be allowed to withdraw from further participation at any point of the study without any penalty or consequence. There will be no financial gain or any other incentive for participation in this study. Each participant will be contacted before initiation of the Delphi process, explaining the study and addressing any questions and concerns. Written consent will be required from each participant at the onset of the study, this will be completed in the Delphi platform as part of the first round of the Delphi process. Ongoing consent will be implied with ongoing participation of the study.

One of the advantages specific to the Delphi process is the anonymized responses of participants. Participants’ individual responses will not be visible to each other and will not be available for review. The Delphi platform facilitates anonymized group feedback for building consensus each round. The final group consensus will be made available for review and publication. Relevant demographic details (without identifiers) and work experience will be published, whilst respecting rights to privacy and confidentiality of participants.

## 3. Discussion

District hospitals form the backbone of healthcare service delivery in the South African context. There are clear guidelines on the structure, staffing and equipment requirements for district hospitals as set out in the South African Ideal Hospital Framework. This specifically includes having ultrasound access in both the casualty and maternity units [29]. Despite having widespread access to ultrasound units, at present there are no official guidelines on what ultrasound examinations should be offered at a South African district hospital. Without clear guidelines on its use, there has not been any standardised training of ultrasound use for medical practitioners working in the district hospital setting, or for family physicians as part of their speciality training. The absence of such guidelines is potentially limiting the scope of service delivery at district hospital level and may directly affect patient care as well as referral pathways.

This study will adopt the Delphi method to achieve expert consensus on which ultrasound skillsets are mandatory and optional within the South African district hospital context. The use of the Delphi method ensures that all individual contributions are anonymized, and that each experts’ contribution is equally weighted. Inclusion of experts drawn from all academic departments of family medicine and district hospitals across the country will ensure adequate representation of the various role players in the district health care delivery. This may enhance greater acceptance and implementation of study findings. Findings from this study may form the core curriculum for ultrasound training of medical practitioners in the district hospitals and registrars in the academic departments of family medicine in South Africa. This study may also provide further guidance on what ultrasound skillsets are not suitable for district hospitals, a finding which may guide referral and management pathways for certain medical conditions. A notable limitation of the proposed study is failure of or delayed response from included participants. If there is a poor uptake or retention of experts in this voluntary Delphi process, it may invalidate the findings of the study. As such, there may be need for enhanced reminders (emails and telephone calls) for the participants to improve response rate.

## 4. Conclusions

This study will create the first set of guidelines on the ultrasound skillsets that should be expected from district-level medical practitioners in South Africa. It will also establish a clear set of skills that patients may expect from medical practitioners whom they consult at district hospitals. With the inclusion of stakeholders from South African medical schools and a geographically diverse group of medical practitioners from across the country, this study will not only shed light on what POCUS skills are required but may create a foundation for further ultrasound curriculum development. The perspectives gained may be used to help formulate registrar training in ultrasound for the College of Family Physicians of South Africa. This in turn will improve the quality and efficiency of district hospital service delivery, with the ultimate consequence of improving healthcare to rural and underserved communities.

## Figures and Tables

**Figure 1 ijerph-19-09640-f001:**
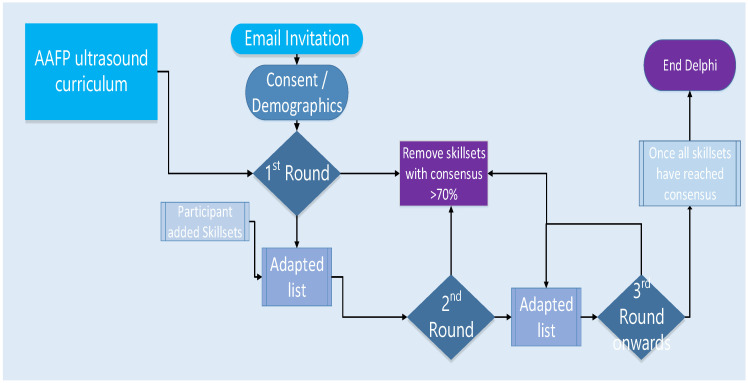
The proposed survey process.

**Figure 2 ijerph-19-09640-f002:**
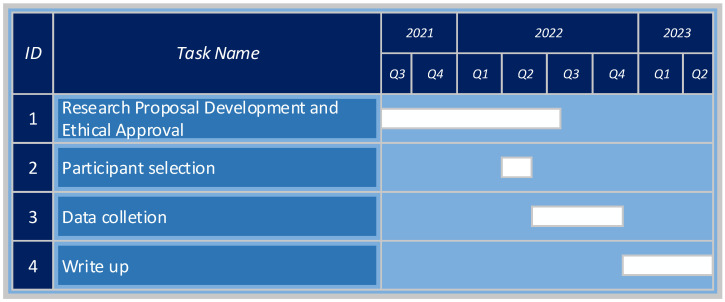
Study Schedule.

**Table 1 ijerph-19-09640-t001:** Selection of experts from academic departments of family medicine (Source: Mash et al. [35]).

Province	University	No.
Western Cape	University of Cape Town (UCT)	2
	Stellenbosch University (US)	2
Eastern Cape	Walter Sisulu University (WSU)	2
KwaZulu-Natal	University of KwaZulu Natal (UKZN)	2
Free State	University of the Free State (UFS)	2
Gauteng	University of Witwatersrand (Wits)	2
	University of Pretoria (UP)	2
	Sefako Makgotho Health Sciences University (SMU)	2
Limpopo	University of Limpopo (UL)	2
	Total	18

## Data Availability

Participant data has not yet been collected. All demographic information on the participants will be password protected and managed as per the South African Protection of Personal Information Act (POPIA) principles. All participant responses will remain anonymous, as per Delphi protocol. Participant responses will be anonymized and made available as addendum to the final published findings.

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
