# Peer review of "Protocol for a Delphi Consensus Study to Determine the Essential and Optional Ultrasound Skills for Medical Practitioners Working in District Hospitals in South Africa"

_ijerph, 2022, doi:10.3390/ijerph19159640_

Round 1

Reviewer 1 Report

The submitted manuscript shows a well-elaborated protocol for a Delphi consensus study to determine the essential and optional ultrasound skills for medical practitioners. The protocol is dedicated to South Africa due to the need to focus attention on the most common pathologies in the reference area and is inspired by the POCUS skillset published by the American Academy of Family Physicians. The product has been well developed and needs minimal revisions.

- add a reference to Table 1.

- add HOD (Head of Department) to the list of abbreviations.

- I suggest adding some sentences about the role of the facilitator and the 4 key features that characterize the Delphi process (www.welphi.com).

- I suggest adding flowcharts for every passage. 

Author Response

Thanks for the prompt review process. Please find below the rebuttal to the queries from the reviewers.

Dr P. Mans (On behalf of the  Authors)

Comment 1: add a reference to Table 1.

Response: We have provided a reference for the table as requested.

Mash R, Steinberg H, Naidoo, M. Updated programmatic learning outcomes for the training of family physicians in South Africa. S Afr Fam Pract. 2021;63(1), a5342. https://doi.org/10.4102/safp.v63i1.5342

Comment 2: add HOD (Head of Department) to the list of abbreviations.

Response: Thanks for the suggestion. We have made the correction as suggested.

Comment 3: I suggest adding some sentences about the role of the facilitator and the 4 key features that characterize the Delphi process (www.welphi.com).

Response: We have addressed these comments in line 147 - 152 and line 189 -190.

The facilitator’s role is to supply anonymised group feedback on each item that did not reach consensus after each round (line 148) and to follow up participants who have not responded within the provided time window (line 190)

If required, a dedicated section describing the facilitators role can be created. More sentences were added detailing the facilitators feedback role (see track changes)

The 4 key features mentioned on www.welphi.com include ‘friendly and attractive interface’ as number 4, which is site specific and not Delphi process specific.

The other three features are all addressed in the study design section. See line 147 regarding anonymity, lines 148-152 regarding controlled feedback and line 189 about the time allowed (4weeks) to respond.

Comment 4: I suggest adding flowcharts for every passage.

Response: Thank you for the suggestion. Since this is the study protocol, and the actual study has not yet commenced, we are not able to predict with certainty the number of rounds that will be required to complete the Delphi process. That is why the flow chart has the wording ‘onwards’, indicating the process will complete as many rounds as required to reach consensus.

Reviewer 2 Report

A good article on applying Delphi technique to create a list of appropriate ultrasound skills using expert consensus

Considerations about the article:

1)    Line 41 and 42 have to be modified: there is no risk of exposure to “radiation” meaning “ionizing radiation” during an MRI procedureDuring a MRI scan, patients are exposed to High Static and Gradient Magnetic Fields and to radio frequency waves.[1]

[1] Abi Bergerscience editor How does it work? Magnetic resonance imaging Journal List BMJ v.324(7328); 2002 Jan 5 PMC1121941 BMJ. 2002 Jan 5; 324(7328): 35. doi: 10.1136/bmj.324.7328.35 PMCID: PMC1121941 PMID: 11777806 https://www.ncbi.nlm.nih.gov/pmc/articles/PMC1121941/

2)    Line 291-293 Where the Supplementary Materials (Table 1: Selection of experts from academic departments of family medicine 292 Figure 1: The proposed survey process Table 2: Study Schedule Participant Questionnaire) are?

3)    At the end how many of the experts involved in the Delphi process completed it?

4)    Which are the main differences between ultrasound competencies for physicians working in district hospitals in South Africa and the POCUS competencies published by the American Academy of Family Physicians?

Author Response

Thanks for the prompt review process. Please find below the rebuttal to the queries from the reviewers.

Dr P. Mans (On behalf of the  Authors)

Comment 1.    Line 41 and 42 have to be modified: there is no risk of exposure to “radiation” meaning “ionizing radiation” during an MRI procedure. During a MRI scan, patients are exposed to High Static and Gradient Magnetic Fields and to radio frequency waves.[1]

[1] Abi Berger, science editor How does it work? Magnetic resonance imaging Journal List BMJ v.324(7328); 2002 Jan 5 PMC1121941 BMJ. 2002 Jan 5; 324(7328): 35. doi: 10.1136/bmj.324.7328.35 PMCID: PMC1121941 PMID: 11777806 https://www.ncbi.nlm.nih.gov/pmc/articles/PMC1121941/

Response: Thanks for this insightful comment. We have re-phrased the sentence accordingly.

Comment 2:   Line 291-293 Where the Supplementary Materials (Table 1: Selection of experts from academic departments of family medicine 292 Figure 1: The proposed survey process Table 2: Study Schedule Participant Questionnaire) are?

Response: These figures have been included.  

Comment 3:  At the end how many of the experts involved in the Delphi process completed it?

Response: This article is the research protocol. We have not yet commenced the research. The aim is to include two experts from each academic Family Medicine Department (18) and two frontline workers from each province (18) with a total of 36 experts in the Delphi process.

Comment 4:   Which are the main differences between ultrasound competencies for physicians working in district hospitals in South Africa and the POCUS competencies published by the American Academy of Family Physicians?

Response 5: This question will be answered once the study is completed – in a separate article describing the findings of our study

This manuscript is a resubmission of an earlier submission. The following is a list of the peer review reports and author responses from that submission.